**Data Availability Statement:** Data were collected and owned by the demographic and health survey authority. Data are available at: https://dhsprogram.

# Prevalence and associated factors of underweight, overweight and obesity among women of reproductive age group in the Maldives: Evidence from a nationally representative study

**Mohammad Rashidul Hashan**[1]*, **Md Fazla Rabbi**[2], **Shams Shabab Haider**[3], **Rajat Das Gupta**[3,4,5]

1 Bangladesh Civil Service, Ministry of Health and Family Welfare, Dhaka, Bangladesh, 2 Bangladesh National Nutrition Council, Health Service Division, Ministry of Health and Family Welfare, Dhaka, Bangladesh, 3 Centre for Science of Implementation & Scale Up, BRAC James P Grant School of Public Health, BRAC University, Mohakhali, Dhaka, Bangladesh, 4 Centre for Non-Communicable Disease and Nutrition, BRAC James P Grant School of Public Health, BRAC University, Mohakhali, Dhaka, Bangladesh, 5 Department of Epidemiology and Biostatistics, Arnold School of Public Health, University of South Carolina, Columbia, South Carolina, United States of America

* hasandmck66@gmail.com

## Abstract

### Background

Global epidemiological transition across various countries have documented the coexistence of undernutrition and overnutrition. South Asian countries are facing this public health hazard in remarkable manner. To enrich the evidence and relation with women's health in the Maldives, this study was undertaken to examine the prevalence and associated factors of underweight, overweight and obesity among reproductive age women.

### Methods

This study was conducted utilizing data from the Maldives Demographic and Health Survey 2016–17. After presenting descriptive analyses, multivariable logistic regression analysis method was used to examine the prevalence and associations between different nutritional status categories. These were grouped based on the WHO recommended cut-off value and relevant socio-demographic determinants among reproductive age women.

### Results

A total weighted sample of 6,634 reproductive age Maldivian women (15–49 years) were included in the analysis. The overall prevalence of overweight and obesity was 63%, while the underweight prevalence was 10%. The younger age group (15–24 years) had a higher prevalence of underweight (26%). On the other hand, an overweight and obesity prevalence of 82.6% was observed among the older age group (35–49 years). Regression analysis showed that residents of the North and Central Provinces, those in the higher quintiles of

com/data/dataset/Maldives_Standard-DHS_2016.
cfm?flag=0. Following instruction, data are
available to download. Anyone interested to work
with these data will be able to access these data in
the same manner as the authors. Without the
permission of DHS authority, the authors cannot
share the de-identified data set.

**Funding:** The author(s) received no specific
funding for this work.

**Competing interests:** The authors have declared
that no competing interests exist.

wealth index, married women and those with parity of more than two children, were all signif-
icantly negatively correlated to being underweight. Increased age, being married or sepa-
rated/divorced/widowed and having more than three children was found to have a
significant positive association with overweight and obesity.

## Conclusions

Maldives is facing nutritional transition and a major public health hazard demonstrated by
the high burden of overweight and obesity and persistence of chronic problem of undernutri-
tion. Surveillance of vulnerable individuals with identified socio-demographic factors and
cost-effective interventions are highly recommended to address the persistent underweight
status and the emerging problem of overweight/obesity.

## Introduction

Overweight/obesity is one of the leading risk factors of developing chronic diseases like cardio-
vascular diseases, cancer, diabetes and chronic respiratory diseases, which account for 71% of
total global deaths [1]. The world has witnessed a three-fold increase in obesity since 1975.
Now, more than half of the total adult population ($\geq$18 years old) are overweight and obese
[2,3]. According to the 2018 Global Nutrition Report, there was a slow decrease in the preva-
lence of underweight women from 11.6% in 2000 to 9.7% in 2016 as well as a sharp increase in
adult obesity of about 0.32 kg/m$^2$ per decade [2,4]. A recent systematic review revealed that the
global prevalence of low body mass index (BMI) was slightly more than high BMI and the
prevalence of underweight was mostly evident in the Asian and African region. Conversely,
the rise of obesity is also predominant in the African and Asian region, while it has remained
stable in high income countries [5]. Concurring with the ongoing epidemiological transition
towards increased prevalence of nationwide overnutrition, undernutrition still prevails as a
major public health problem in low and middle income countries (LMICs) [6–8]. There have
been several studies reporting the persistence of this dual-edged sword of malnutrition across
different strata of individual, household, city and country levels [5,9–11].

Underweight and overweight threaten both an individual's survival and a health system's
resilience [5]. The overwhelming effect of this double-edged sword reduces human productiv-
ity and results in an economic catastrophe [12,13]. This is specially important for women of
reproductive age group. For example, maternal BMI is associated with pregnancy outcome. A
systematic review showed that a slight increase in a mother's BMI was associated with
increased risk of adverse pregnancy outcomes like maternal mortality, fetal death, stillbirth,
neonatal death, perinatal death, infant death and development of respiratory diseases of the
children [14,15]. Another study showed the positive association of maternal underweight and
fetal growth retardation, death of neonates and stunting in case of survivors [5].

Underweight and overweight among reproductive age women has been shown to be influ-
enced by age, socio-economic status (SES), educational status, area of residence, marital status
etc. In earlier studies, increased age had been found to be associated with the BMI of women
in various magnitudes [10,16–20]. Furthermore, SES was also found to be a significant factor
of underweight and overweight among women as this correlated to the availability of junk
food and high energy yielding food [10,16,18,19]. Women with more education were less likely
to be involved performing physical activities and thus to be underweight and more likely to be
overweight [17,18]. Area of residence is also an important factor in determining the weight of

women [10,16,20]. Different dietary practices among adolescent girls were found to be responsible for being underweight of adult women [21]. Watching television was also found to be associated with obesity among reproductive age women [11,22]. In addition, variation in stature, family size, pregnancy or marital status and parity were all found to be associated with the nutrition status of women in different countries [10,17,19]. Different studies have aimed at analyzing prevalence and some factors of female nutritional status [23–25]. However, to the best of our knowledge, this study is the first comprehensive attempt to analyze the nutritional status of women of the Maldives.

The Maldives Demographic and Health Survey (MDHS) 2016–2017 collected information on nutritional status and relevant factors of reproductive age women. These data have provided an opportunity to analyze the weight contrast of reproductive age women. In this study, we aim to evaluate the prevalence and associated factors of underweight, overweight and obesity of reproductive age women.

## Methods

### Study settings and data source

This study is the secondary analysis of cross-sectional data from MDHS 2016–17. The MDHS 2016–17 is a nationally representative survey which collected data from March 2016 to November 2017 [26]. It covered information on key demographic characteristics and health indicators such as family planning, maternal and children health, nutritional status, non-communicable disease, domestic violence and fertility data. The work was implemented by the Ministry of Health (MoH) with financial support from the Government of the Maldives, WHO, UNICEF, UNFPA and technical assistance from ICF International (USA) thorough the DHS program. The sampling frame of the survey was taken from the Maldives Population and Housing Census 2014. Thereafter, the participants in MDHS 2016–17 were selected using probability proportion based on two-stage stratified cluster sampling from the chosen sampling frame. In the initial stage, sample clusters were selected from the main sampling frame of the 2014 census. In the second stage, an equal systematic sample of 42 households was selected from each cluster; 6,697 households in total. The detailed protocol and methods were published previously [25]. This study extracted data from the woman's questionnaire, which was used to collect information from all women aged 15–49 years. In brief, 9,170 women were approached with a response rate of 84%. In this study, we excluded pregnant women and those women who delivered two months prior to data collection.

### Data collection and measurements

Trained health staff collected anthropometric data of weight and height from interviewed participants utilizing calibrated measurement tools. A lightweight SECA with digital screen, UNICEF electronic scales set on a flat surface were used for weight and height measurement. The measuring board had an accuracy of ±0.1 cm and ±0.1 kg, respectively. Asian specific BMI cut-off criteria was used to categorize underweight ($<18.5$ kg/m$^2$), normal weight ($\geq 18.5$ kg/m$^2$ to $<23$ kg/m$^2$), overweight and obesity ($\geq 23$ kg/m$^2$ to $<27.5$ kg/m$^2$).

The outcome variable of the study was the participants' BMI, which indicated their nutritional status. Comprehensive literature review was performed to select the following relevant independent variables for this study: age (15–24, 25–34, 35–49 years); place of residence (rural, urban); regions (Malé, North, North Central, Central, South, South Central); educational status (no formal education, primary, secondary, higher); current employment status (no, yes); wealth status (poorest, poorer, middle, richer, richest); marital status (single, married, separated/divorced/widowed); parity (0, 1, 2, 3, $\geq 4$ children); number of household member ($\leq 5$,

>5 individuals); and frequency of watching television (not at all, less than once a week, at least once a week). Wealth index of the participant was calculated using principle component analysis of household assets, including electronics, bicycles, sanitation facilities, water sources, use of health services, etc. The calculated index was then divided into quintiles. All of the survey information was collected from participants during a face-to-face interview using a pre-tested questionnaire.

## Data analysis

Descriptive analysis was performed to report the frequencies and percentages of selected co-variates on the background characteristics of the participants based on BMI status. The prevalence of each category was estimated, along with the overall prevalence of each socio-demographic characteristic. These background co-variates were selected based on published literature and available data from MDHS 2016–17. Multivariable logistic regression analysis was done to obtain the association of each independent variable with the dependent variable (i.e. underweight and overweight/obesity), utilizing a normal BMI range as the reference value. Crude and adjusted regression models were built and variables with a pre-specified significance value of <0.2 in the unadjusted model were eligible for inclusion in the final adjusted multivariable models [27]. Association results of multivariable regression analysis were presented by odds ratio (OR) at 95% confidence intervals (CIs). Statistical significance was considered with $p$-value <0.05.

## Ethics approval

The survey protocol, including biomarker collection, was reviewed and approved by the National Health Research Committee of the Maldives and the Institutional Review Board of ICF. Participants agreed voluntarily to take part in the survey through written informed consent. We received approval to utilize the dataset for secondary analysis from DHS on May 2020.

## Findings

Table 1 demonstrates the results of the descriptive analysis of BMI categories from 6,634 Maldivian women of reproductive age (15–49 years old) in regard to various socio-economic characteristics. It shows that the overall prevalence of overweight and obesity was 63%, which is almost six times higher compared to the prevalence of underweight (10%, p <0.0001). The highest prevalence (26%) of underweight women belonged to youngest age group (15–24 years) while the middle age group (25–34 years) had the lowest prevalence (2.5%). Whereas the prevalence of overweight and obesity was highest (82.6%) among the oldest group of women (35–49 years), the lowest prevalence (36.7%) was among the youngest age group (15–24 years). In both urban (61.4%) and rural (66.6%) areas, respondents had an almost five times higher proportion of overweight and obesity compared to the proportion of underweight (10% and 12%, respectively; p <0.01). Around three out of five women were overweight or obese in every region with the highest proportion in South Central (69.5%) and North Central (67.5%) regions. Women with secondary education had the highest prevalence (15.5%) of underweight status while overweight and obesity prevalence increased as educational level decreased with the highest proportion (82.9%) among women with only primary education (p <0.001). The prevalence of overweight and obesity was higher among currently married women (75.4%) compared to single (32.6%) and separated/divorced/widowed (70.1%) women. At the same time, the prevalence of underweight was higher among the single women (38.4%), compared to currently married (20.1%) and separated/divorced/widowed (22.6%) women. Finally,

**Table 1. Distribution of basic characteristics of respondents according to BMI status, n (%)\* (N = 6,634).**

| Variable | Frequency (N = 6634) | Percentage (%) | BMI Status (%) | | | p-value |
|---|---|---|---|---|---|---|
| | | | Underweight (n = 719) | Normal Weight (n = 1645) | Overweight and Obese (n = 4270) | |
| **Age Group (years)** | | | | | | |
| 15–24 | 2025 | 30.5 | 26.0 | 37.3 | 36.7 | <0.0001 |
| 25–34 | 2238 | 33.7 | 5.9 | 24.0 | 70.1 | |
| 35–49 | 2371 | 35.7 | 2.5 | 14.9 | 82.6 | |
| **Place of Residence** | | | | | | |
| Urban | 2880 | 43.4 | 12.0 | 26.6 | 61.4 | 0.0122 |
| Rural | 3754 | 56.6 | 10.0 | 23.4 | 66.6 | |
| **Regions** | | | | | | |
| Malé | 2880 | 43.4 | 12.0 | 26.6 | 61.4 | 0.0074 |
| North Region | 878 | 13.2 | 10.7 | 24.7 | 64.6 | |
| North Central | 834 | 12.6 | 9.7 | 22.8 | 67.5 | |
| Central Region | 393 | 5.9 | 7.5 | 28.2 | 64.3 | |
| South Central | 746 | 11.3 | 10.0 | 20.6 | 69.5 | |
| South Region | 904 | 13.6 | 10.6 | 23.1 | 66.2 | |
| **Highest Educational Status** | | | | | | |
| No Formal Education | 290 | 4.4 | 3.3 | 16.1 | 80.7 | <0.0001 |
| Primary | 1545 | 23.3 | 3.2 | 14.0 | 82.9 | |
| Secondary | 3450 | 52.0 | 15.5 | 27.2 | 57.2 | |
| Higher | 1351 | 20.4 | 9.3 | 32.8 | 58.0 | |
| **Currently Employed** | | | | | | |
| No | 3866 | 58.3 | 12.3 | 24.1 | 63.6 | 0.0046 |
| Yes | 2769 | 41.7 | 8.8 | 25.8 | 65.4 | |
| **Wealth index** | | | | | | |
| Poorest | 1220 | 18.4 | 11.4 | 23.4 | 65.1 | 0.1566 |
| Poorer | 1272 | 19.2 | 10.3 | 23.1 | 66.6 | |
| Middle | 1347 | 20.3 | 10.9 | 24.6 | 64.6 | |
| Richer | 1387 | 20.9 | 13.2 | 25.1 | 61.7 | |
| Richest | 1408 | 21.2 | 8.5 | 27.4 | 64.1 | |
| **Marital Status** | | | | | | |
| Single | 1634 | 24.6 | 29.0 | 38.4 | 32.6 | <0.0001 |
| Currently Married | 4412 | 66.5 | 4.6 | 20.1 | 75.4 | |
| Separated/Divorced/Widowed | 588 | 8.9 | 7.2 | 22.6 | 70.1 | |
| **Parity** | | | | | | |
| 0 | 2349 | 35.4 | 23.6 | 35.4 | 41.1 | <0.0001 |
| 1 | 1248 | 18.8 | 6.1 | 25.1 | 68.9 | |
| 2 | 1298 | 19.6 | 3.1 | 19.6 | 77.3 | |
| 3 | 821 | 12.4 | 3.5 | 15.4 | 81.2 | |
| >3 | 918 | 13.8 | 2.5 | 13.1 | 84.4 | |
| **Number of Household Member** | | | | | | |
| ≤5 | 2652 | 40.0 | 8.1 | 24.4 | 67.6 | <0.0001 |
| >5 | 3982 | 60.0 | 12.7 | 25.1 | 62.2 | |
| **Frequency of Watching Television** | | | | | | |
| Not at all | 443 | 6.7 | 39.5 | 33.2 | 27.3 | 0.1163 |
| Less than once a week | 480 | 7.2 | 31.1 | 32.1 | 36.8 | |

*(Continued)*

**Table 1.** (Continued)

| Variable | Frequency (N = 6634) | Percentage (%) | BMI Status (%) | | | p-value |
|---|---|---|---|---|---|---|
| | | | Underweight (n = 719) | Normal Weight (n = 1645) | Overweight and Obese (n = 4270) | |
| At least once a week | 5711 | 86.1 | 35.7 | 31.8 | 32.5 | |

*Column percentage.

nulliparous women had a higher proportion (23.6%) of underweight status. The overweight and obese proportion successively increased with increased parity compared to underweight status (p <0.001).

## Determinants of underweight

Table 2 illustrates multinomial logistic regression analysis to show association estimates for underweight participants compared to normal weight for the explored socio-demographic co-variates. Participants who resided in the North and Central regions, were in the higher wealth index quintiles, were married and who had a parity of more than two children were identified as inversely correlated to being underweight compared to the normal weight individuals across those variables in the analyzed sample and this finding was statistically significant. Furthermore, the number of household members showed a significant positive association with being underweight relative to normal weight participants.

Women from every region of the Maldives had a significantly lower likelihood of being underweight compared to the Malé region, with the most reduced odds being among Central Region residents (OR = 0.4; 95% CI 0.2–0.7, p <0.001). Married women reduced their risk of being underweight to half (OR = 0.5; 95% CI 0.3–0.7, p <0.001) that of an unmarried individual. Increase in wealth index tended to be inversely associated with being underweight, as respondents from the richest quintile had a 70% reduced chance of being underweight (OR = 0.3; 95% CI 0.2–0.6, p <0.001) relative to the poorest wealth quintile. However, no such association was observed in the case of overweight/obesity for wealth index status. There was significantly decreased risk (OR = 0.5; 95% CI 0.3–0.9, p <0.02) of developing underweight among women as they increased parity. Parity of more than 3 children made an individual 1.4 times more likely to be overweight or obese (OR = 1.4; 95% CI 1.0–1.9, p <0.02) relative to nulliparous women.

## Determinants of overweight and obesity

Table 3 shows the multinomial logistic regression analysis for association estimates comparing overweight and obese to normal weighted individuals for the background characteristics. Increased age, being married or separated/divorced/widowed and having more than three children were all found to have significant positive association with overweight and obesity relative to normal weighted individuals in this study. Women from the 25–34 year age group are 1.4 times more likely to be overweight or obese (OR = 1.4; 95% CI 1.2–1.8, p <0.001). This likelihood increases with age, such that women from the 35–49 year age group were 1.8 times more at risk compared to women of the youngest age group (15–24 years). Married women's chances of being overweight or obese increased to more than double (OR = 2.4; 95% CI 1.9–3.0, p <0.001), while the odds of women who are separated/divorced/widowed increased 1.7 times (OR = 1.7; 95% CI 1.3–2.3, p <0.001) compared to unmarried individuals. Women living in a household consisting of more than five members had 1.4 times greater odds (OR = 1.4;

**Table 2. Crude and adjusted odds ratio (95% CI) estimates of underweight compared to normal weight by respondent background characteristics.**

| Variable | Crude Odds Ratio (COR) (95% Confidence Interval) | Adjusted Odds Ratio (AOR) (95% Confidence Interval) |
|---|---|---|
| **Age Group (in years)** | | |
| 15–24 | Ref | Ref |
| 25–34 | 0.4 (0.3–0.5)*** | 0.9 (0.7–1.3) |
| 35–49 | 0.2 (0.2–0.3)*** | 0.6 (0.4–1.0) |
| **Place of Residence** | | |
| Rural | Ref | |
| Urban | 1.1 (0.9–1.5) | |
| **Regions** | | |
| Malé | Ref | Ref |
| North Region | 0.9 (0.7–1.3) | 0.6 (0.4–1.0)* |
| North Central | 0.9 (0.7–1.3) | 0.6 (0.3–0.9)* |
| Central Region | 0.6 (0.4–0.9)** | 0.4 (0.2–0.7)*** |
| South Central | 1.0 (0.7–1.4) | 0.7 (0.4–1.1) |
| South Region | 0.9 (0.7–1.3) | 0.6 (0.4–1.0)* |
| **Highest Educational Status** | | |
| No Formal Education | Ref | Ref |
| Primary | 1.1 (0.6–2.2) | 1.0 (0.5–2.1) |
| Secondary | 2.5 (1.3–4.7)** | 0.8 (0.4–1.8) |
| Higher | 1.5 (0.8–0.3) | 0.6 (0.3–1.4) |
| **Currently Employed** | | |
| No | Ref | Ref |
| Yes | 0.7 (0.6–0.8)*** | 0.8 (0.7–1.0) |
| **Wealth index** | | |
| Poorest | Ref | Ref |
| Poorer | 0.8 (0.7–1.1) | 0.8 (0.6–1.0) |
| Middle | 0.8 (0.6–1.0)* | 0.7 (0.5–0.9)** |
| Richer | 0.8 (0.6–1.1) | 0.5 (0.3–0.8)** |
| Rich | 0.6 (0.4–1.0)* | 0.3 (0.2–0.6)*** |
| **Marital Status** | | |
| Single | Ref | Ref |
| Married | 0.3 (0.2–0.3)*** | 0.5 (0.3–0.7)*** |
| Separated/Divorced/Widowed | 0.4 (0.3–0.6)*** | 0.6 (0.4–1.0)* |
| **Parity** | | |
| 0 | Ref | Ref |
| 1 | 0.4 (0.3–0.6)*** | 0.8 (0.5–1.2) |
| 2 | 0.3 (0.2–0.4)*** | 0.5 (0.3–0.8)** |
| 3 | 0.2 (0.1–0.4)*** | 0.5 (0.3–0.8)** |
| >3 | 0.2 (0.2–0.4)*** | 0.5 (0.3–0.9)* |
| **Number of Household Member** | | |
| ≤5 | Ref | Ref |
| >5 | 1.4 (1.1–1.7)*** | 1.4 (1.1–1.7)** |
| **Frequency of Watching Television** | | |
| Not at all | Ref | Not included in the final model |

(*Continued*)

**Table 2.** (Continued)

| Variable | Crude Odds Ratio (COR) (95% Confidence Interval) | Adjusted Odds Ratio (AOR) (95% Confidence Interval) |
|---|---|---|
| Less than once a week | 0.8 (0.5–1.3) | |
| At least once a week | 1.0 (0.7–1.4) | |

Note-

*p-value<0.05

**p-value<0.01

***p-value<0.00.

95% CI 1.1–1.7, p <0.004) of being underweight compared to those that had less than five household members. Place of residence, educational status, employment status and frequency of watching television did not reveal any association with any of the BMI categories.

## Discussion

This present study, to the best of our knowledge, is the first examination of the nationwide prevalence and factors associated with overweight/obesity and underweight among women of reproductive age from the Maldives. To conduct such estimates, nationally representative data from MDHS 2016–17 was utilized. Categorization was done using the Asia-specific BMI cut-off criteria. Our results demonstrate the overall prevalence of overweight/obesity (64.3%), normal weight (24.7%) and underweight (10%). Findings from this study also illustrate various socio-demographic factors including regions of residence, wealth index, marital status and parity as significant correlates of being underweight compared to normal weight individuals whereas only age, marital status and parity of more than three children was found to be associated with being overweight and obesity.

This study shows overweight/obesity prevalence is considerably higher, and on the rise (from 46% in 2009 to 64.3%). There is also an increase in the prevalence of underweight (8% in 2009 to 10%) among women of reproductive age; although, it is much lower compared to overweight/obesity [28]. Such findings reiterate predictions that the prevalence of overnutrition will exceed undernutrition by 2015 and further validates the co-existence of overweight/obesity and underweight within the regional demography; known as "double burden" of malnutrition [29,30]. These findings are consistent with several countries from South Asia and Africa [31–34]. The overnutrition predominance of the Maldivian population reflects the ongoing global nutrition transition. Besides, more than three out of every five women from the Maldives were overweight/obese in this analyzed sample, which correlates with a similar pattern of higher BMI among women that was also reported in Global Burden of Disease studies [35]. Similar findings were reported in studies of women with higher BMI conducted in Asia and Africa [36–41].

The prevalence of underweight (10%) was almost similar to a study done in China (7.8%) which utilized a large sample (16,742,344 women aged 20 to 49 years and 178,556 girls aged 15 to 19 years). However, the prevalence of overweight and obesity in Maldives (63%) was much higher compared to that study. This difference may be due to the difference in sample size, geographical context, and environmental factors between Maldives and China [42].

Current studies show that rural women had a slightly higher prevalence of overweight/obesity in comparison to urban women. However, neither urban nor rural residence was associated with being underweight and overweight/obesity. This is inconsistent with previous studies where urban women had a higher predisposition of being overweight/obese due to

**Table 3. Crude and adjusted odds ratio (95% CI) estimates of overweight/obesity compared to normal weight by respondent background characteristics.**

| Variable | Crude Odds Ratio (COR) (95% Confidence Interval) | Adjusted Odds Ratio (AOR) (95% Confidence Interval) |
|---|---|---|
| **Age Group (in years)** | | |
| 15–24 | Ref | Ref |
| 25–34 | 2.7 (2.3–3.1)*** | 1.4 (1.2–1.8)*** |
| 35–49 | 4.8 (4.1–5.6)*** | 1.8 (1.4–2.4)*** |
| **Place of Residence** | | |
| Rural | Ref | |
| Urban | 0.8 (0.6–0.9)* | |
| **Regions** | | |
| Malé | Ref | Ref |
| North Region | 1.1 (0.8–1.4) | 0.8 (0.6–1.2) |
| North Central | 1.3 (1.0–1.6)* | 1.0 (0.7–1.5) |
| Central Region | 1.0 (0.7–1.2) | 0.8 (0.5–1.1) |
| South Central | 1.4 (1.1–1.8)*** | 1.2 (0.8–1.7) |
| South Region | 1.1 (0.9–1.4) | 1.0 (0.7–1.4) |
| **Highest Educational Status** | | |
| No Formal Education | Ref | Ref |
| Primary | 1.1 (0.8–1.6) | 1.2 (0.8–1.6) |
| Secondary | 0.4 (0.3–0.5)*** | 0.8 (0.6–1.2) |
| Higher | 0.4 (0.3–0.6)*** | 0.7 (0.5–1.1) |
| **Currently Employed** | | |
| No | Ref | Ref |
| Yes | 1.0 (0.9–1.2) | 1.0 (0.9–1.2) |
| **Wealth index** | | |
| Poorest | Ref | Ref |
| Poorer | 0.9 (0.8–1.1) | 0.9 (0.7–1.1) |
| Middle | 1.0 (0.8–1.1) | 1.0 (0.8–1.2) |
| Richer | 0.9 (0.7–1.1) | 1.0 (0.8–1.3) |
| Rich | 0.8 (0.6–1.0) | 1.0 (0.7–1.6) |
| **Marital Status** | | |
| Single | Ref | Ref |
| Married | 4.5 (3.8–5.2)*** | 2.4 (1.9–3.0)*** |
| Separated/Divorced/ Widowed | 3.2 (2.5–4.1)*** | 1.7 (1.3–2.3)*** |
| **Parity** | | |
| 0 | Ref | Ref |
| 1 | 2.1 (1.8–2.5)*** | 0.9 (0.7–1.2) |
| 2 | 3.0 (2.5–3.5)*** | 1.1 (0.8–1.4) |
| 3 | 3.7 (3.0–4.5)*** | 1.2 (0.9–1.6) |
| >3 | 5.0 (4.1–6.1)*** | 1.4 (1.0–1.9)* |
| **Number of Household Member** | | |
| ≤5 | Ref | Ref |
| >5 | 0.8 (0.7–0.9)** | 0.9 (0.8–1.0) |
| **Frequency of Watching Television** | | |
| Not at all | Ref | Not included in the final model |

*(Continued)*

**Table 3.** (Continued)

| Variable | Crude Odds Ratio (COR) (95% Confidence Interval) | Adjusted Odds Ratio (AOR) (95% Confidence Interval) |
|---|---|---|
| Less than once a week | 1.2 (0.9–1.5) | |
| At least once a week | 1.1 (0.9–1.4) | |

Note-

*p-value<0.05

**p-value<0.01

***p-value<0.001.

urbanization and sedentary lifestyles, consumption of energy rich food, decline in physical activity, etc. [21,43–45]. Further explorative study is needed to understand why the prevalence of overweight/obesity was higher in the rural area compared to the urban area in Maldives. To ensure the accomplishment of the sustainable development goals, health promotional programs need to be implemented equitably across regions and vulnerable groups to address any potential nutritional burden [46]. The findings of this study revealed older women are more prone to overweight/obesity compared to their younger counterparts. Similar outcomes were also reported in many other developing countries, suggesting that with increasing age, women tend to reduce their physical activity level and gradually increase their intake of energy dense foods [18,31,32,43]. Also, there is an uneven increase in fat mass among women over 30 years of age [47,48].

This study found that when compared to an unmarried individual, ever married women had a significantly higher likelihood of being overweight/obese and a significantly lower likelihood of being underweight. This finding is in line with previous research conducted in different countries [32,49,50]. Ever married women in our sample, at the time of the survey, may have been from the older age group, which could explain such results. It has also been suggested that widowed/separated women may be more inclined to stay indoors and have less access to physical exercise or outdoor activities due to social and cultural contexts [31,51]. Reduced physical activity results in less energy expenditure, which subsequently increases body weight and fat mass. These factors combined predispose an individual to be overweight/obese [52].

The wealth index of women was found to be a significant predictor of being underweight in the analyzed sample. It showed an inverse relationship of being underweight with the women belonging to the upper wealth quintiles. However, no such statistical significance was observed for being overweight or obese when wealth index was considered. In many developing countries, with increases in income, there are also increases in expenditure and propensity towards sedentary living, obesogenic food consumption and use of technology and modern transportation for convenience [53–55]. All of these behaviors increase one's likelihood of becoming overweight or obese. However, in our study, we could not find any significant relationship between overweight and obesity and the wealth quintile covariate. Nevertheless, being in the higher wealth index tier reduced the likelihood of being underweight, which bolsters the impact of economic solvency in mitigating the burden of undernourishment, as reported in neighboring countries [56–58]. Perhaps, other factors like pattern of age, education level, physical activity, dietary habit or genetic influence overrode the effect of wealth quintile in predicting the risk of being overweight/obese in our sample population. Women with a higher number of children are less likely to be underweight and having more than three children came with a 40% increased risk of developing overweight/obesity compared with women who

had no children. Successive pregnancies cause cumulative excess weight gain during each consecutive pregnancy period as well as during the post-partum timeline as reported in several studies [59–61]. A previous study reported parity as an independent predictor for subsequent maternal weight gain [62].

## Strengths and limitations

To date, this is the first analysis report from a nationally representative dataset exploring multiple modifiable and non-modifiable variables regarding women of reproductive age from the Maldives. It utilized an Asia-specific cut-off point for BMI categories to provide an estimate of nutritional status for this group of people. However, due to the cross-sectional nature of the data, caution needs to be taken when interpreting the data to infer causal relationship. The lack of important covariate data, such as, pattern of physical activity, dietary habit and family history might have introduced confounding biases in our multinomial model analysis. Although BMI is an important WHO recommended indicator of nutritional status measurement, categorizing with such a tool might misclassify the individual data. BMI cannot differentiate body fat and lean body mass. Waist circumference and waist-to-hip ratio could better reflect abdominal obesity.

## Conclusion

This study's findings confirmed that the nutritional transition occurring with the higher burden of overweight and obesity and the persistence of undernutrition. Every three Maldivian women of reproductive age out of five was found to be overweight or obese in the analyzed sample. Surveillance of vulnerable individuals with the identified socio-demographic risk factors and cost-effective interventions are highly recommended to address the persistent underweight status and the emerging problem of overweight/obesity. Further studies are needed to explore dietary habits, physical activity adherences and potential chronic diseases and risk factors that may be contributing to this public health problem.

## Acknowledgments

We are grateful to the DHS program for providing access to the dataset.

## Author Contributions

**Conceptualization:** Mohammad Rashidul Hashan, Rajat Das Gupta.

**Data curation:** Mohammad Rashidul Hashan.

**Formal analysis:** Mohammad Rashidul Hashan, Rajat Das Gupta.

**Methodology:** Mohammad Rashidul Hashan, Rajat Das Gupta.

**Software:** Mohammad Rashidul Hashan, Shams Shabab Haider, Rajat Das Gupta.

**Supervision:** Rajat Das Gupta.

**Validation:** Mohammad Rashidul Hashan.

**Visualization:** Mohammad Rashidul Hashan, Md Fazla Rabbi.

**Writing – original draft:** Mohammad Rashidul Hashan, Md Fazla Rabbi, Shams Shabab Haider.

**Writing – review & editing:** Mohammad Rashidul Hashan, Md Fazla Rabbi, Shams Shabab Haider, Rajat Das Gupta.

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
