## [Decision Letter · Decision Letter 0]

7 Sep 2020

PONE-D-20-21827

Prevalence and associated factors of underweight, overweight and obesity among women of reproductive age group in the Maldives: Evidence from a nationally representative study.

PLOS ONE

Dear Dr. Hasan,

Thank you for submitting your manuscript to PLOS ONE. After careful consideration, we feel that it has merit but does not fully meet PLOS ONE’s publication criteria as it currently stands. Therefore, we invite you to submit a revised version of the manuscript that addresses the points raised during the review process.

We look forward to receiving your revised manuscript.

Kind regards,

Russell Kabir, PhD

Academic Editor

PLOS ONE

Journal Requirements:

Reviewers' comments:

Reviewer's Responses to Questions

**Comments to the Author**

1. Is the manuscript technically sound, and do the data support the conclusions?

Reviewer #1: Partly

Reviewer #2: Yes

2. Has the statistical analysis been performed appropriately and rigorously? 

Reviewer #1: No

Reviewer #2: Yes

3. Have the authors made all data underlying the findings in their manuscript fully available?

Reviewer #1: No

Reviewer #2: Yes

4. Is the manuscript presented in an intelligible fashion and written in standard English?

Reviewer #1: Yes

Reviewer #2: Yes

5. Review Comments to the Author

Reviewer #1: Comments to authors

I am happy by reviewing this manuscript. For improving the quality of the paper, the following comments were forwarded for the authors.

Title: Prevalence and associated factors of underweight, overweight and obesity among women of the reproductive age group in the Maldives: Evidence from a nationally representative study

1. In the abstract, methods subsection, the author said ‘’Secondary analysis was performed to present descriptive statistics.’’ Make it clear???.

2. In the introduction section; you stated “Watching television was also found to be associated with obesity among reproductive-age women (11)” make it explain in terms of frequency, it is crucial ?? , add more references to this? For support, look at this paper “Mohammed Ahmed, Abdu Seid , and Adnan Kemal. Does the Frequency of Watching Television Matters on Overweight and Obesity among Reproductive Age Women in Ethiopia?. Journal of Obesity. Volume 2020, Article ID 9173075, 7 pages “

3. What makes women different from men in experiencing thus nutritional outcomes. Please state in the introduction appropriately?

4. In the Methods section the author included women of reproductive age ? are you include/ exclude pregnant and postpartum women?

5. In the Methods section, the author states “Overweight and obesity was combined into one category for descriptive purposes.”. Why you merge this outcome in the descriptive?. This is a mistake(misclassification of outcomes)? Therefore, what is the value of the WHO cut off point?

6. In the selection of your variables, Why did not include alcohol drinking status, contraceptive history, and frequency of watching television among the women? This is an important variable

7. Multi-collinearity among the independent variables included in the model was assessed VIF. Does it advisable for a logistic regression analysis?

8. In the descriptive statistics, you reported p-value, from what type of statistic do you get? ( from what type of chi-square, since it is weighted sample due to two-stage stratified cluster sampling)

9. I have a serious concern in your analysis, what type of logistic regression analysis utilized ? is it binary logistic /multinomial logistic?. Besides, How many categories of outcome do you have?

10. Line 184 states “significantly negatively correlated to being underweight compared to the normal weight” reworded it?

11. In the discussion, What is your justification being rural women had a higher prevalence of overweight/obesity in comparison to urban women? Please state it well?

12. Line 284. Harris et al. reported parity as a 285 independent predictor for subsequent maternal weight gain (61). remove Harris et al?

Reviewer #2: The authors have addressed an important issue regarding studying vital physical conditions women of reproductive age group in the Maldives. Overall, the manuscript is well written, however, many sentences should be simplified rather representing through complex sentences.

The data was collected from 6634 individual women and was analyzed in a reasonable way. In the discussion section, there should be some sentences explaining if there are other factors also related for the nutritional transition in the Maldives, such as environmental changes. I would suggest the authors to include a paragraph showing similarity and differences with a coherent published study based on Chinese women and Adolescent girls, titled: "Prevalence of Underweight, Overweight, and Obesity Among Reproductive-Age Women and Adolescent Girls in Rural China." There, the research team compared and studied far more larger population data (16 742 344 women aged 20 to 49 years and 178 556 girls aged 15 to 19) and due to geological and environmental differences there might be some interesting comparison.

6. PLOS authors have the option to publish the peer review history of their article (what does this mean?). If published, this will include your full peer review and any attached files.

Reviewer #1: No

Reviewer #2: **Yes: **Dr Ehsanul Hoque Apu, DDS. MSc. PhD.

---

## [Author Response · Author response to Decision Letter 0]

25 Sep 2020

PONE-D-20-21827

Prevalence and associated factors of underweight, overweight and obesity among women of reproductive age group in the Maldives: Evidence from a nationally representative study.

Reviewer #1: 

I am happy by reviewing this manuscript. For improving the quality of the paper, the following comments were forwarded for the authors.

Title: Prevalence and associated factors of underweight, overweight and obesity among women of the reproductive age group in the Maldives: Evidence from a nationally representative study

1. In the abstract, methods subsection, the author said ‘’Secondary analysis was performed to present descriptive statistics.’’ Make it clear???.

Response: Thanks, we have revised the statement like following: “After presenting descriptive analyses, multivariable logistic regression analysis method was used to examine the prevalence and associations between different nutritional status categories.”

2. In the introduction section; you stated “Watching television was also found to be associated with obesity among reproductive-age women (11)” make it explain in terms of frequency, it is crucial ?? , add more references to this? For support, look at this paper “Mohammed Ahmed, Abdu Seid , and Adnan Kemal. Does the Frequency of Watching Television Matters on Overweight and Obesity among Reproductive Age Women in Ethiopia?. Journal of Obesity. Volume 2020, Article ID 9173075, 7 pages “

Response: Thank you for this comment. We have added the reference as per suggestion.

3. What makes women different from men in experiencing thus nutritional outcomes. Please state in the introduction appropriately?

Response: Thank you. We mentioned in the introduction: “Underweight and overweight threaten both an individual’s survival and a health system’s resilience (5). The overwhelming effect of this double-edged sword reduces human productivity and results in an economic catastrophe (12,13). This is specially important for women of reproductive age group. For example, maternal BMI is associated with pregnancy outcome. A systematic review showed that a slight increase in a mother’s BMI was associated with increased risk of adverse pregnancy outcomes like maternal mortality, fetal death, stillbirth, neonatal death, perinatal death, infant death and development of respiratory diseases of the children (14,15). Another study showed the positive association of maternal underweight and fetal growth retardation, death of neonates and stunting in case of survivors (5).”

4. In the Methods section the author included women of reproductive age ? are you include/ exclude pregnant and postpartum women?

Response: Thank you for this important comment. Yes, we excluded the pregnant and postpartum women. We mentioned that in the revised manuscript: “In this study, we excluded pregnant women and those women who delivered two months prior to data collection.”

5. In the Methods section, the author states “Overweight and obesity was combined into one category for descriptive purposes.”. Why you merge this outcome in the descriptive?. This is a mistake(misclassification of outcomes)? Therefore, what is the value of the WHO cut off point?

Response: Thanks! We removed the sentence. However, we analyzed overweight and obesity as a combined category based on literature. This is because our interest is to prevent underweight or excessive weight. So, we combined overweight and obesity to find out the factors associated with overweight and obesity in comparison to normal weight.

We followed these articles based on DHS data. All of them combined overweight and obesity as a single category:

• Al Kibria GM, Swasey K, Hasan MZ, Sharmeen A, Day B. Prevalence and factors associated with underweight, overweight and obesity among women of reproductive age in India. Global health research and policy. 2019 Dec 1;4(1):24. URL: https://ghrp.biomedcentral.com/articles/10.1186/s41256-019-0117-z

• Biswas T, Garnett SP, Pervin S, Rawal LB. The prevalence of underweight, overweight and obesity in Bangladeshi adults: Data from a national survey. PloS one. 2017 May 16;12(5):e0177395. URL: https://journals.plos.org/plosone/article?id=10.1371/journal.pone.0177395

• Rawal LB, Kanda K, Mahumud RA, Joshi D, Mehata S, Shrestha N, Poudel P, Karki S, Renzaho A. Prevalence of underweight, overweight and obesity and their associated risk factors in Nepalese adults: data from a Nationwide Survey, 2016. PloS one. 2018 Nov 6;13(11):e0205912. URL: https://www.ncbi.nlm.nih.gov/pmc/articles/PMC6219769/

6. In the selection of your variables, Why did not include alcohol drinking status, contraceptive history, and frequency of watching television among the women? This is an important variable.

Response: Thank you for this important comment. We have included frequency of watching television among the women in the revised model. Data on alcohol drinking status was not collected. Contraceptive history was present only among married women, while our sample included both married and unmarried women. 

7. Multi-collinearity among the independent variables included in the model was assessed VIF. Does it advisable for a logistic regression analysis?

Response: Thanks! We have removed this sentence. 

8. In the descriptive statistics, you reported p-value, from what type of statistic do you get? ( from what type of chi-square, since it is weighted sample due to two-stage stratified cluster sampling)

Response: Thanks! The p-value was obtained from corrected weighted Pearson chi square statistic. 

9. I have a serious concern in your analysis, what type of logistic regression analysis utilized ? is it binary logistic /multinomial logistic?. Besides, How many categories of outcome do you have?

Response: Thanks! it was multivariable logistics regression. Using normal weight as the reference category, multivariable logistic regression analyses were conducted to investigate the associated factors of underweight and combined overweight/obesity. There were three categories of outcome: underweight (˂18.5 kg/m2), normal weight (≥18.5 kg/m2 to ˂23 kg/m2), overweight and obesity (≥23 kg/m2).

10. Line 184 states “significantly negatively correlated to being underweight compared to the normal weight” reworded it?

Response: Thanks! We have reworded it as per following: “Participants who resided in the North and Central regions, were in the higher wealth index quintiles, were married and who had a parity of more than two children were identified as inversely correlated to being underweight compared to the normal weight individuals across those variables in the analyzed sample and this finding was statistically significant.”

11. In the discussion, What is your justification being rural women had a higher prevalence of overweight/obesity in comparison to urban women? Please state it well?

Response: Thanks! We have revised the statement as such: “Current studies show that rural women had a slightly higher prevalence of overweight/obesity in comparison to urban women. However, neither urban nor rural residence was associated with being underweight and overweight/obesity. This is inconsistent with previous studies where urban women had a higher predisposition of being overweight/obese due to urbanization and sedentary lifestyles, consumption of energy rich food, decline in physical activity, etc.. Further explorative study is needed to understand why the prevalence of overweight/obesity was higher in the rural area compared to the urban area in Maldives.”

12. Line 284. Harris et al. reported parity as a 285 independent predictor for subsequent maternal weight gain (61). remove Harris et al?

Response: Thank you! We have removed Harris et al. 

Reviewer #2: 

The authors have addressed an important issue regarding studying vital physical conditions women of reproductive age group in the Maldives. Overall, the manuscript is well written, however, many sentences should be simplified rather representing through complex sentences.

The data was collected from 6634 individual women and was analyzed in a reasonable way. In the discussion section, there should be some sentences explaining if there are other factors also related for the nutritional transition in the Maldives, such as environmental changes. I would suggest the authors to include a paragraph showing similarity and differences with a coherent published study based on Chinese women and Adolescent girls, titled: "Prevalence of Underweight, Overweight, and Obesity Among Reproductive-Age Women and Adolescent Girls in Rural China." There, the research team compared and studied far more larger population data (16 742 344 women aged 20 to 49 years and 178 556 girls aged 15 to 19) and due to geological and environmental differences there might be some interesting comparison.

Response: Thank you for this important comment. We have added the following statement: “The prevalence of underweight (10%) was almost similar to a study done in China (7.8%) which utilized a large sample (16,742,344 women aged 20 to 49 years and 178,556 girls aged 15 to 19 years). However, the prevalence of overweight and obesity in Maldives (63%) was much higher compared to that study. This difference may be due to the difference in sample size, geographical context, and environmental factors between Maldives and China.”

---

## [Editor Report · Decision Letter 1]

19 Oct 2020

Prevalence and associated factors of underweight, overweight and obesity among women of reproductive age group in the Maldives: Evidence from a nationally representative study.

PONE-D-20-21827R1

Dear Dr. Hassan,

We’re pleased to inform you that your manuscript has been judged scientifically suitable for publication and will be formally accepted for publication once it meets all outstanding technical requirements.

Kind regards,

Russell Kabir, PhD

Academic Editor

PLOS ONE
---

## [Editor Report · Acceptance letter]

21 Oct 2020

PONE-D-20-21827R1 

Prevalence and associated factors of underweight, overweight and obesity among women of reproductive age group in the Maldives: Evidence from a nationally representative study 

Dear Dr. Hashan:

I'm pleased to inform you that your manuscript has been deemed suitable for publication in PLOS ONE. Congratulations! Your manuscript is now with our production department. 

Kind regards, 

on behalf of

Dr. Russell Kabir 

Academic Editor

PLOS ONE